# Broadband Achromatic Metalens in the Visible Light Spectrum Based on Fresnel Zone Spatial Multiplexing

**DOI:** 10.3390/nano12234298

**Published:** 2022-12-03

**Authors:** Ruixue Shi, Shuling Hu, Chuanqi Sun, Bin Wang, Qingzhong Cai

**Affiliations:** 1School of Instrumentation and Optoelectronics Engineering, Beihang University, Beijing 100191, China; 2Institute of Microelectronics of the Chinese Academy of Sciences, Beijing 100029, China

**Keywords:** achromatic broadband metalens, Fresnel zone spatial multiplexing, particle swarm algorithm

## Abstract

Metalenses composed of a large number of subwavelength nanostructures provide the possibility for the miniaturization and integration of the optical system. Broadband polarization-insensitive achromatic metalenses in the visible light spectrum have attracted researchers because of their wide applications in optical integrated imaging. This paper proposes a polarization-insensitive achromatic metalens operating over a continuous bandwidth from 470 nm to 700 nm. The silicon nitride nanopillars of 488 nm and 632.8 nm are interleaved by Fresnel zone spatial multiplexing method, and the particle swarm algorithm is used to optimize the phase compensation. The maximum time-bandwidth product in the phase library is 17.63. The designed focal length can be maintained in the visible light range from 470 nm to 700 nm. The average focusing efficiency reaches 31.71%. The metalens can achieve broadband achromatization using only one shape of nanopillar, which is simple in design and easy to fabricate. The proposed metalens is expected to play an important role in microscopic imaging, cameras, and other fields.

## 1. Introduction

Optical lenses are the most basic element of optical system. Lenses have been widely used in security, smart phones, and other fields. In the 21st century, with the rapid development of optical technology, the integration, miniaturization, and reduction in weight of optical systems have become the most urgent needs at present. Traditional lens uses the principle of phase accumulation [1] to obtain the expected wavefront of the beam passing through it; the thickness of the lens increases with diameter. In order to achieve high-quality imaging, the chromatic aberration of lens imaging must be solved by using lens groups or aspheric lenses [2,3]. This increases the difficulty of optical instruments integration.

A metasurface composed of a large number of subwavelength nanostructures has the ability to manipulate the incident wavefront. It manipulates the phase, polarization, and amplitude of light by precisely tailoring the geometry and rotation of nanostructures [4,5,6,7,8]. Therefore, metasurfaces are used to develop various miniature optical devices, such as metalenses [9,10,11,12,13,14,15,16,17,18,19,20,21,22,23,24,25,26,27,28,29,30,31,32,33,34,35,36,37,38], holograms [39,40,41], waveplates [42,43], and optical vortices [44,45]. Metalenses are one of the most important applications of metasurfaces. A metalens focuses the light at a certain point via the phase shifters when the light scatters off the array of resonators making up the metalens [9]. Compared with a traditional lens, a metalens is smaller and the control of the light field is more flexible, which provides the possibility for the miniaturization and integration of the optical system. There are numerous applications based on metalenses, such as light field imaging [46,47], color routing [48], orbital angular momentum multiplexing [49], and fiber focusing [50].

Achromatic metalenses are one of the most researched types of metalenses [18,19,20,21,22,23,24,25,26,27,28,29,30,31,32,33,34,35,36,37,38]. Metalenses suffer from chromatic aberration, which reduces the focal length as the incident wavelength increases, resulting in blurring and color distortion. Correcting chromatic aberration in the broadband region is key to achieving full-color imaging. Designing broadband achromatic metalenses is a great challenge as it is difficult to construct the phase profiles of different wavelengths on a metasurface. The cascading and spatially multiplexing methods can only achieve discrete wavelengths of achromatic metalenses [19,20,21,22]. Continuous broadband achromatic metalenses have been realized using the geometric phase [23,24,25,26,27]. A pair of nanofins as a coupled phase-shift unit can be applied to manipulate the phase, group delay, and group delay dispersion of light, increasing the degree of freedom of design [23]. A metalens array can capture light-field information and achieve a full-color light-field camera which is made of gallium nitride (GaN) nanoantenna and complementary GaN nanoholes, providing base phase and phase compensation [28]. However, these design methods have a shortage of polarization sensitivity. Using a variety of unit cells with different cross-sections is another approach. Different shapes of nanostructures should be carefully selected by linear fitting so that the effective refractive index does not vary with frequency [29,30]. However, these methods of constructing a large phase library with complex cross-sections limit the product of time delay and spectral bandwidth and pose fabrication challenges.

To our knowledge, few designs have applied Fresnel zone multiplexing to achieve broadband achromatic metalenses. The dense vertical stacking of three independent metalenses has been introduced to achieve achromatic design at three wavelengths [19]. Multiplexing two Fresnel zones radially can be used to focus both the left circularly polarized light and the right circularly polarized light [10]. Diffraction-limited achromatic metalenses have been demonstrated by exploiting the constructive interference of light from multiple zones and dispersion engineering [32], which is polarization-sensitive.

Broadband polarization-insensitive achromatic metalenses in the visible light spectrum have attracted researchers because of their wide applications in optical integrated imaging [46,47]. In this paper, a polarization-insensitive achromatic metalens in the visible light spectrum using the Fresnel zone spatial multiplexing method and the particle swarm optimization (PSO) algorithm is proposed. Silicon nitride nanopillars are selected as the building blocks of the metalens. The phase library provides a large product of the time delay and bandwidth. The nanostructures of 488 nm and 632.8 nm are alternately arranged by Fresnel zone spatial multiplexing and the PSO algorithm is used to optimize the phase compensation, realizing a dual-wavelength achromatic design. A focal length of 20 μm can be maintained in the range of 470–700 nm with a focal length variation of 7.26%, satisfying the broadband achromatic design. The proposed method introduces only one nanostructure shape to reduce chromatic aberrations and make fabrication easier.

## 2. Design Principle and Methods

Metalenses focus plane waves to a tiny spot. The phase profile of a metalens must satisfy the following equation [36]:(1)$$\varphi (x,y,z,\lambda )=-\frac{2\pi }{\lambda }(\sqrt{x^{2}+y^{2}+f^{2}}-f)+C(\lambda ),$$
where (*x*, *y*) is the position from the center of the metalens, *f* is the focal length, and *λ* is the target wavelength. *C*(*λ*) is introduced as an additional phase and is a constant. *C*(*λ*) can take any value depending on the incident wavelength and be applied as a free parameter in the design process. According to Equation (1), the phase profile is obtained with different profiles for different wavelengths. The achromatic metalens is designed with a diameter (*d*) of 14 μm and a focal length (*f*) of 20 μm. The numerical aperture of the metalens is 0.33, which is defined as *NA* = sin[arctan(*d*/2*f*)].

The metalens plane is discretized into several −π to π periods. The meta-unit is filled into each period to provide the effective refractive index and meet the requirements of phase shift and transmittance. The deviation between the target and the realized phase profile can be minimized via tuning *C*(*λ*).

Intelligent optimization algorithms can be used to select the appropriate *C*(*λ*) and the desired nanostructures. The dual-wavelength Fresnel zone spatial multiplexing method and the particle swarm optimization algorithm are applied to achieve a polarization-insensitive achromatic metalens in the range of 470 nm to 700 nm. Firstly, the meta-units of different sizes are swept to obtain the phase, transmittance and the product of time delay and spectral bandwidth to build the phase library. Secondly, the desired phase profile is obtained according to Equation (1). The metalens surface at 488 nm and 632.8 nm is discretized, respectively. Thirdly, the designed metalens surface is divided into four zones radially. Using Fresnel zone spatial multiplexing, the meta-units of the two individual metalenses at 488 nm and 632.8 nm are interleaved in the four zones. Then, the PSO algorithm is used to tune *C*(*λ*) to minimize the total wave deviation.

### 2.1. Phase and Transmittance of the Unit Cell

There are two ways to modulate the unit cell: the geometric phase and the propagation phase modulation. The geometric phase process adjusts the phase by rotating the angle of the meta-unit. Unlike the geometric phase, the propagation phase modulation is determined by changing the duty cycle of the unit cell and is polarization-insensitive. Silicon nitride has high transmittance in the visible light spectrum and is easy to integrate. Therefore, the silicon nitride nanopillar with the propagation phase is discussed in this paper.

Figure 1a shows the 3D view of the unit cell. The unit cell consists of a Si_3_N_4_ nanopillar deposited on the glass substrate. Figure 1b shows the top view of the different unit cells. The period *P*, diameter *D*, and height *H* of the nanopillar must be precisely designed to achieve the propagation phase coverage from −π to π.

The nanopillar represents a truncated waveguide and the required phase is achieved as waveguiding effects [36]. To satisfy the Nyquist sampling criterion, the period of the nanopillar should be less than *λ*/2*NA* [11]. Additionally, the period should be smaller than the incident wavelength to suppress higher-order diffraction [36,51,52]. For continuous broadband achromatic metalenses, the period should be larger than the wavelength band of the incident light to excite the guided mode resonance to provide more phases [36,51,52]. The higher the unit cell, the larger the phase modulation range that can be achieved, but a higher nanopillar makes fabrication much more difficult. The period was fixed at 380 nm, and the height was 1.3 μm. The nanopillar can be fabricated by plasma-enhanced chemical vapor deposition and inductively coupled with a plasma reactive ion etching process [33].

The diameter was swept from 50 to 380 nm via the finite difference time domain method. In the simulation, the *x*-polarized plane light was incident from the structures base, the periodic boundary conditions were set in the *x* and *y* directions, and the perfectly matched layer was set in the *z* direction. Figure 1c shows the magnetic field distributions inside the nanopillar with a diameter of 120 nm at 488 nm and 632.8 nm, respectively. The nanopillar acts like a low-quality factor Fabry–Perot resonator and confines most of the light [11]. Therefore, the coupling effect between adjacent unit cells can be ignored. The phase and transmittance resonance of 488 nm and 632.8 nm are shown in Figure 1d,e, respectively. The phase modulation was achieved by changing the diameter of the nanopillar, and the phase was wrapped in −π to π. As shown in Figure 1c,d, expect for the falling peaks, the transmittance is high. The falling peaks are produced by guided mode resonances when the incident light is coupled with the surface mode of the periodic lattice [49,50]. In the simulation and fabrication of metalenses, the nanopillars corresponding to the falling peaks were removed so that each unit cell in the phase library maintained a high transmittance. Multiple −π to π periods are realized, which helps find the appropriate meta-units when designing the achromatic metalens.

### 2.2. Time-Bandwidth Product of the Phase Library

As mentioned above, each meta-unit can be considered as a truncated waveguide based on the equivalent medium theory. Equation (1) can be written as [50]
(2)$$\varphi (\omega ,f)=\frac{\omega }{c}n_{eff}(\omega )H,$$
where *ω* is the angular frequency of incident light, *c* is the speed of light in a vacuum, and *n_eff_*(*ω*) represents the effective refractive index. The corresponding phases of the nanopillars at different frequencies were obtained by simulation. Then, *n_eff_*(*ω*) can be calculated by Equation (2). The group velocity ∂*ω*/∂*k* can be derived as
(3)$$\frac{\partial \omega }{\partial k}=\frac{c}{n_{eff}+\omega \frac{dn_{eff}}{d\omega }},$$
where *k* is the wavenumber: *k* = *n_eff_ω*/*c*. The time delay *T* is
(4)$$T=H(\frac{1}{\nu_{g}}-\frac{1}{c})=H(\frac{n_{eff}(\omega )+\omega \frac{dn_{eff}}{d\omega }-1}{c}).$$

In the 470–700 nm waveband, the spectral bandwidth (∆ω) is 1.32 × 10^15^ rad/s. The effective refractive index is linear with the frequency. Thus,
(5)$$\frac{dn_{eff}}{d\omega }=\frac{n_{eff}^{\max}-n_{eff}^{\min}}{\omega_{\max}-\omega_{\min}}=\frac{n_{eff}^{\max}-n_{eff}^{\min}}{\Delta \omega },$$
(6)$$T=H(\frac{n_{eff}(\omega )+\omega \frac{n_{eff}^{\max}-n_{eff}^{\min}}{\Delta \omega }-1}{c}).$$

The center frequency (*ω_o_* = 1.11 × 10^15^ rad/s) is selected. The time-bandwidth product (TBP) is
(7)$$T\Delta \omega =H\Delta \omega (\frac{n_{eff}(\omega_{o})+\omega_{o}\frac{n_{eff}^{\max}-n_{eff}^{\min}}{\Delta \omega }-1}{c}).$$

Figure 2 shows the time-bandwidth products corresponding to different diameters of meta-units in the phase library. (*T*∆*ω*)*_library_* is defined as the maximum TBP in the phase library, which is 17.63.

The time-bandwidth product of a metalens must be smaller than the TBP of the phase library. The time delay of the designed metalens is [53]
(8)$$T=\frac{\sqrt{f^{2}+\frac{1}{4}d^{2}}-f}{c}.$$

The bandwidth of the metalens which can be achieved can be calculated as
(9)$${\left(\Delta \omega \right)}_{\max}\leq \frac{{\left(\Delta T\Delta \omega \right)}_{library}}{T}.$$

In this paper, the time delay of the metalens is 3.97 fs. Therefore, the maximum bandwidth that can be achieved is 4.44 × 10^15^ rad/s, which is greater than the bandwidth 1.32 × 10^15^ rad/s we are interested in. This proves that the phase library meets our requirements.

### 2.3. Fresnel Zone Spatial Multiplexing Metalens Design

For a single-wavelength metalens design, the target phase corresponding to each position is calculated by Equation (1), and the unit cell providing the realized phase with the smallest deviation from the target phase is found from the phase library. The unit cells corresponding to 488 nm and 632.8 nm are combined to achieve a dual-wavelength achromatic metalens. Motivated by the spatial multiplexing method [10], we considered the possibility of segmenting the lens into different zones along the radial direction and using the PSO algorithm to arrange the unit cells in order to obtain achromatic design [32,34,36,54]. The electric field at the focal spot is determined by the interference of the electric field within each zone and the interference of the electric fields from different zones. Therefore, it is necessary to reasonably select the number and the radius of Fresnel zones, as well as the unit cells of each zone, so as to maximize the electric field intensity at the focal point. First, we attempted to separate the metalens into two zones; however this was unsuccessful due to the presence of bi-focus (Appendix A). We then sought to make it into four zones with a range of 1:1:1:1 but the focal length at 632.8 nm did not meet our requirements (Appendix A). Finally, we set the zone’s range to 3:2:1:1 (Appendix A), which met our requirements.

Figure 3a,b show the top and perspective view of the metalens, respectively. *O* is the center of the metalens, and the metalens is divided into four zones along the radial direction. The four zones have radii *R*_1_, *R*_2_, *R*_3_, and *R*_4_ in the ratio of 3: 2: 1: 1. According to the Fresnel zone spatial multiplexing method, the nanostructures at 488 nm and 632.8 nm are interleaved. Zones I and III correspond to the unit cells of 488 nm (blue), and zones II and IV correspond to the unit cells of 632.8 nm (red). The PSO algorithm is used to adjust C(*λ*) and arrange the unit cells of the metalens (Appendix A).

For a nanopillar, its phase response to different wavelengths is different. A definite metasurface phase profile struggles to satisfy the achromatic requirements of two or more wavelengths. The additional phase, *C*(*λ*), can provide phase compensation, which is the key to achieving an achromatic metalens. Therefore, different wavelengths have different phases at the center to match the given phase profile. The PSO algorithm is used to optimize the *C*(*λ*) corresponding to 488 nm and 632.8 nm, respectively, and to arrange the unit cells by zone multiplexing throughout the whole design process of the metalens. *C*(*λ*) is the position of the particle, and the deviation is used as the fitness function. The total deviation is defined as
(10)$$\Delta =\sum_{\left(x,y\right)}\sum_{\lambda_{i}}\left|\varphi_{t}-\varphi_{r}\right|,$$
where *φ_t_* is the target phase and *φ_r_* is the realized phase in the phase library. The total phase deviation is the sum of the phase deviations of the two wavelengths. The detailed optimization process is shown in Figure 4. In each iteration, C(λ) as the position of the particle swarm is updated and optimized and the unit cell is selected at each (x,y) of the metalens according to Fresnel zone multiplexing. The total deviation is calculated. The iteration of PSO algorithm is used to find the minimum value of total deviation (Δ). When Δ satisfies the Rayleigh criterion, the iteration stops. According to the Rayleigh criterion, when the deviation between the actual wavefront and the ideal wavefront does not exceed *λ*/4, the actual wavefront can be considered defect-free. However, even if the deviation exceeds *λ*/4, when it is small enough, the wavefront can be considered defect-free [55]. In this paper, the number of iterations was 800.

## 3. Results and Discussion

### 3.1. Achromatic Focusing Analysis at 488 nm and 632.8 nm

Figure 5a,b show the normalized light intensity distributions in the *x-z* plane at 488 nm and 632.8 nm, respectively. The incident light of 488 nm and 632.8 nm is focused at 20.28 μm and 20.82 μm, respectively, and the average deviation with the design focal length of 20 μm is 2.25%, which satisfies the requirements of the dual-wavelength achromatic design.

Figure 5c,d show the normalized intensity distributions at the focal plane, and the cross-sectional intensities are shown in Figure 5e,f. The focal points at different wavelengths are Gaussian-distributed circular spots. The focal spot’s full width at half maximum (FWHM) can characterize the focusing quality. The FWMH at 488 nm and 632.8 nm is 603 nm and 904 nm, respectively. Focusing efficiency is defined as the ratio of the optical field energy within a circular area with a radius of three times FWHM to the total incident energy. The focusing efficiency at 488 nm and 632.8 nm is 30.61% and 39.72%, respectively.

### 3.2. Continuous Waveband Achromatic Focusing Analysis at 470–700 nm

After considering the dual-wavelength focusing, the focusing over the whole band was analyzed. The 470–700 nm band is divided into 12 discrete wavelengths. Figure 6a–c shows the normalized intensity profiles in the *x-z* plane, the normalized intensity distributions in the focal plane, and the cross-sectional intensity distributions, respectively. For these 12 wavelengths, the depth of focus (*DOF* = *λ*/*NA*^2^) ranges from 4.35 μm to 6.43 μm, and the maximum value of the focal length variation is 0.82 μm. The depth of focus is larger than the fluctuation range of focal length, indicating that different wavelengths of light are focused on the same point. Table 1 shows the focal length and focusing efficiency of different wavelengths. As a comparison, a chromatic metalens at 488 nm was designed. Figure 7a shows the focal length of the achromatic and chromatic metalens. In the wavelength range of 470–700 nm, the focal length of achromatic metalens hardly changes, whereas the chromatic metalens varies greatly. The focal length change of the achromatic metalens is defined as
(11)$$\sigma =\frac{\max(f)-\min(f)}{\mathrm{mean}(f)},$$
where *max*(*f*) is the maximum focal length, *min*(*f*) is the minimum focal length, and *mean*(*f*) is the average focal length. The calculated variation is 7.26%, which satisfies the achromatic design. In this paper, the time-bandwidth product (T∆ω)_lens_ is 5.22.

Figure 7b shows the focusing efficiency of different wavelengths. The average focusing efficiency is 31.71%. When the incoming light is greater than 550 nm, the focusing efficiency increases with the incident wavelength, reaching a maximum of 47.13% at 700 nm. In general, the focusing efficiency of a metalens does not exceed 50%. Finding methods to improve focusing efficiency is the focus of our subsequent research. Table 2 shows the key parameters of the achromatic metalens. We summarized the studies of achromatic metalenses in the visible band as shown in Table 3.

## 4. Conclusions

In summary, a polarization-insensitive broadband achromatic metalens in the visible region was designed and demonstrated. With silicon nitride nanopillars, the nanostructures of 488 nm and 632.8 nm were alternatively arranged by Fresnel zone spatial multiplexing method and particle swarm optimization algorithm. The upper bound of the time-bandwidth product in the phase library is 17.63. Both 488 nm and 632.8 nm light are focused on the designed focal length with an average focusing efficiency of 35.17%. Meanwhile, the achromatic metalens has a stable focal length in the visible light band of 470–700 nm, with a focal length variation of 7.26%. The average focusing efficiency is 31.71% and the maximum focusing efficiency is 47.13%. The time-bandwidth product of the metalens is 5.22. The nanostructures in the proposed strategy are simple and easy to implement. Our approach provides a reference for other metasurface device designs. It is expected to be applied to many fields, such as cameras, microscopic imaging, and AR/VR devices.

## Figures and Tables

**Figure 1 nanomaterials-12-04298-f001:**
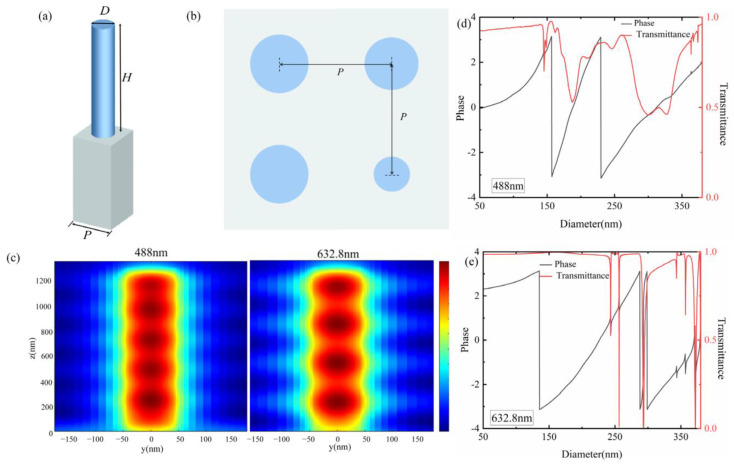
(**a**) 3D view of the unit cells; (**b**) top view of the unit cells; (**c**) the magnetic field distribution inside the nanopillar with a diameter of 120 nm; (**d**) the phase and transmittance of the unit cell of 488 nm; (**e**) the phase and transmittance of the unit cell of 632.8 nm.

**Figure 2 nanomaterials-12-04298-f002:**
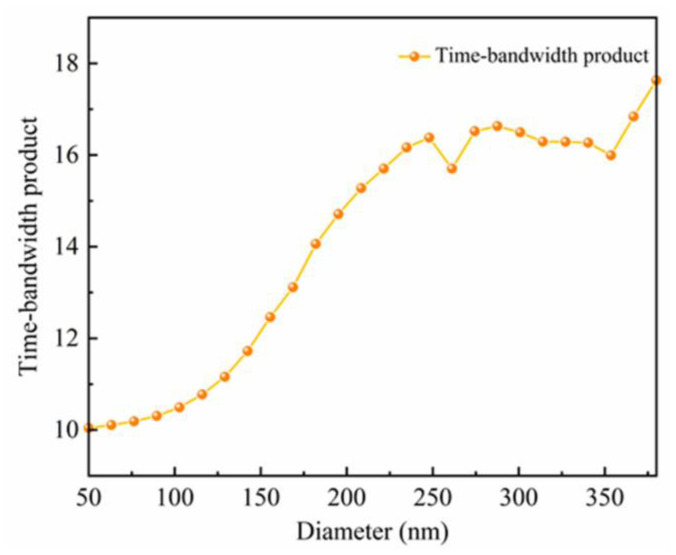
Time-bandwidth products of the meta-units in the phase library.

**Figure 3 nanomaterials-12-04298-f003:**
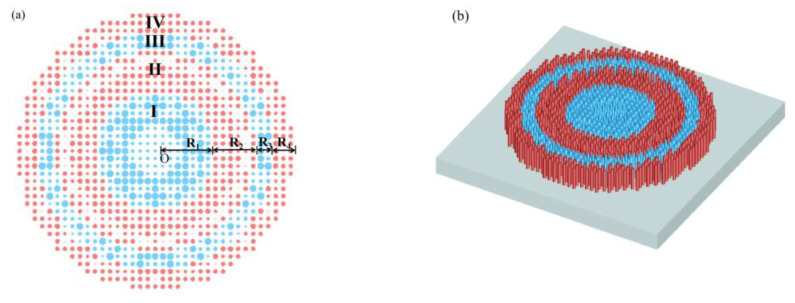
(**a**) Schematic diagram of the dual-wavelength achromatic metalens; (**b**) top view of the designed metalens.

**Figure 4 nanomaterials-12-04298-f004:**
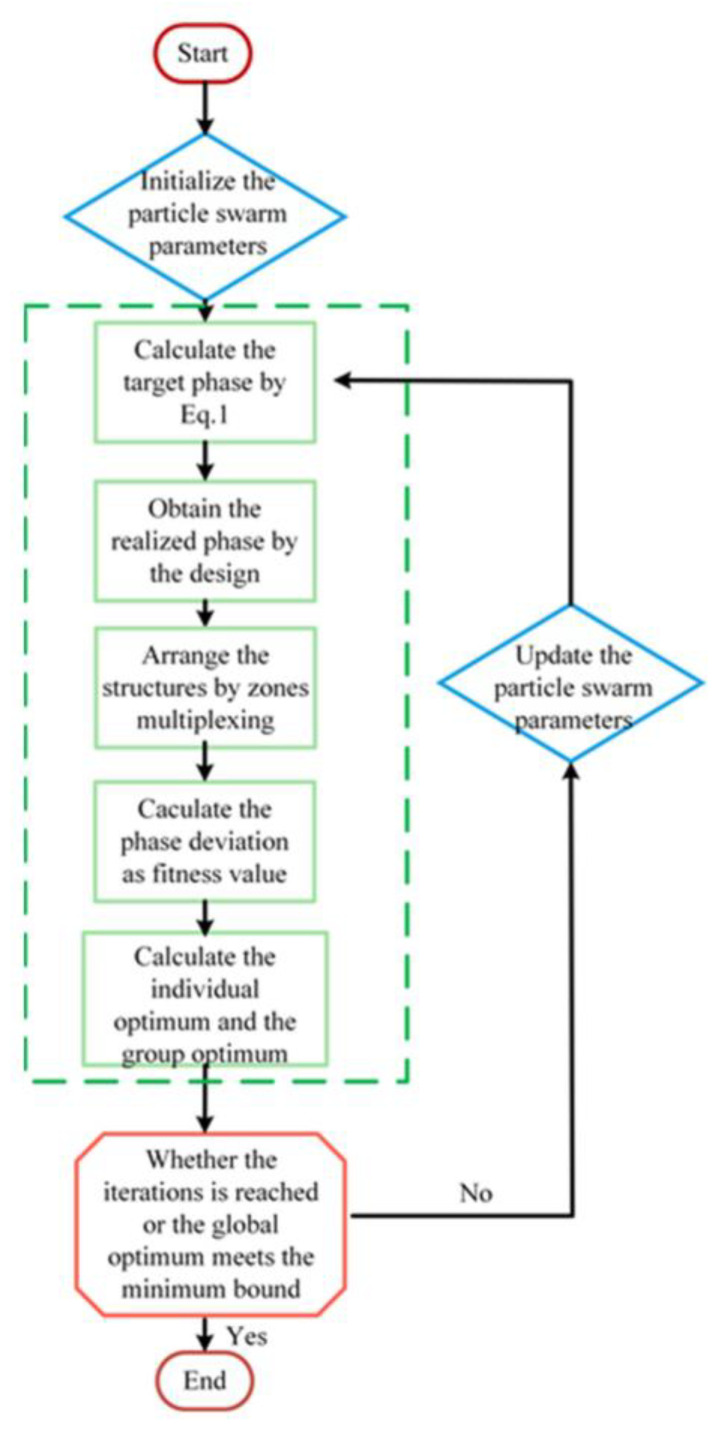
Schematic diagram of the particle swarm algorithm flow.

**Figure 5 nanomaterials-12-04298-f005:**
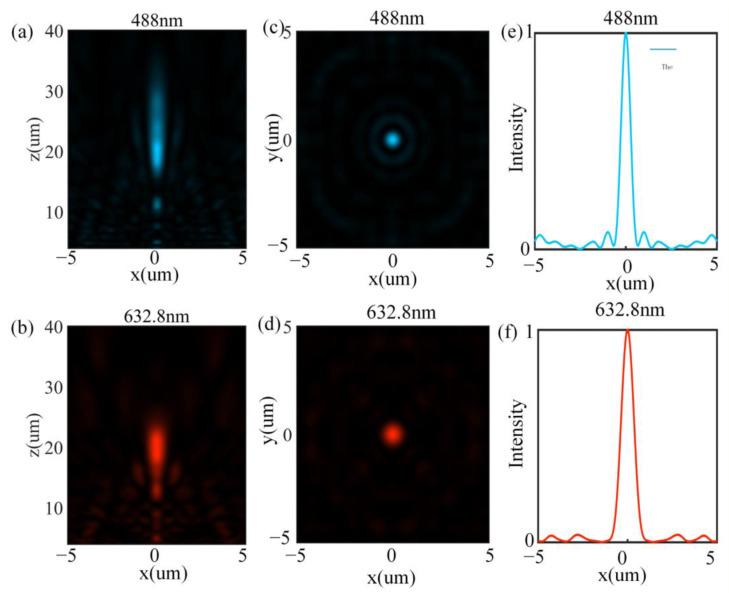
(**a**) Normalized intensity distributions in the *x-z* plane at 488 nm; (**b**) normalized intensity distributions in the *x-z* plane at 632.8 nm; (**c**) normalized intensity distributions of focal spots in the *x-y* plane at 488 nm; (**d**) normalized intensity distributions of focal spots in the *x-y* plane at 632.8 nm; (**e**) the corresponding cross-sectional intensity cut through the focal plane at 488 nm; (**f**) the corresponding cross-sectional intensity cut through the focal plane at 632.8 nm.

**Figure 6 nanomaterials-12-04298-f006:**
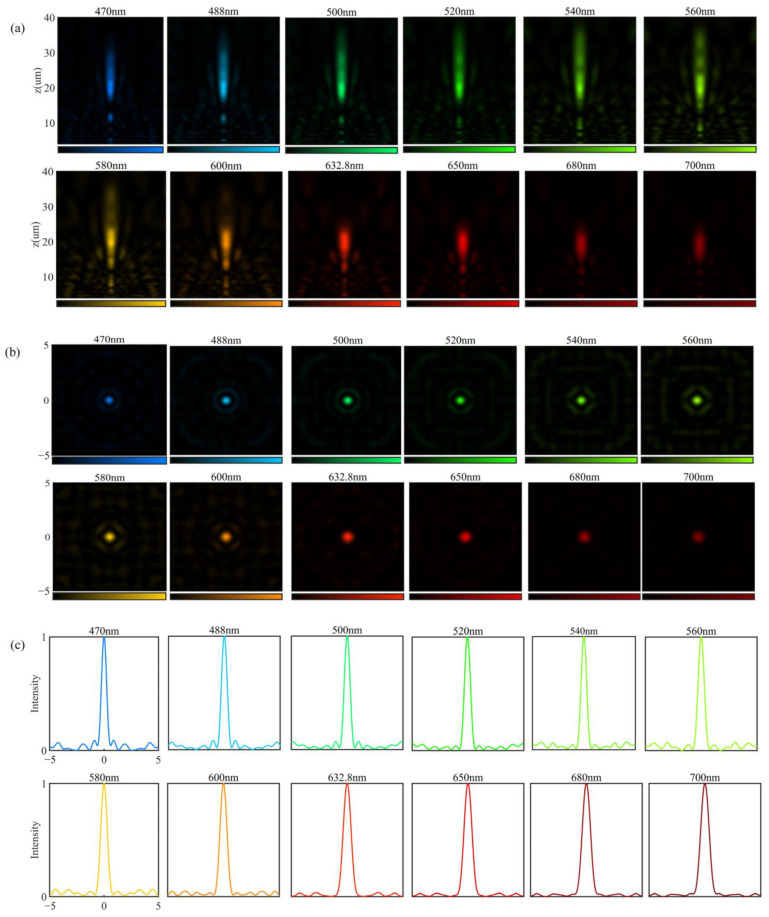
(**a**) Normalized intensity distributions in the *x-z* plane of different wavelengths; (**b**) normalized intensity distributions of focal spots in the *x-y* plane of different wavelengths; (**c**) the corresponding cross-sectional intensity cut through the focal planes of different wavelengths.

**Figure 7 nanomaterials-12-04298-f007:**
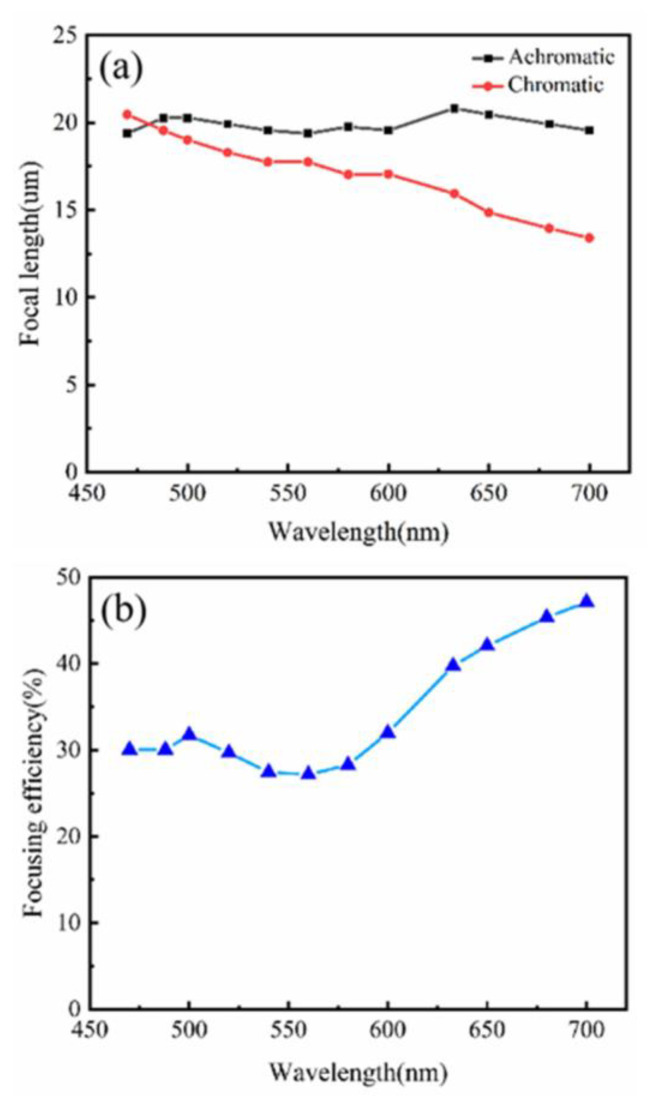
(**a**) Focal length for the chromatic and achromatic metalens of 470–700 nm; (**b**) focusing efficiency for the achromatic metalens of 470–700 nm.

**Table 1 nanomaterials-12-04298-t001:** Focal length and focusing efficiency at different wavelengths of the metalens.

Wavelength	470 nm	488 nm	500 nm	520 nm	540 nm	560 nm	580 nm	600 nm	632.8 nm	650 nm	680 nm	700 nm
Focal length (μm)	19.38	20.28	20.28	19.92	19.56	19.38	19.76	19.56	20.82	20.46	19.92	19.56
Focusing efficiency	30.04%	30.61%	31.74%	29.67%	27.44%	27.17%	28.27%	31.97%	39.72%	42.07%	45.35%	47.13%

**Table 2 nanomaterials-12-04298-t002:** Key parameters of the metalens.

*f* (μm)	*d* (μm)	∆λ (nm)	Average Focusing Efficiency	(*T*∆*ω*)*_library_*	(*T*∆*ω*)*_lens_*	*DOF* (μm)
20	14	470–700	31.71%	17.63	5.22	4.35–6.43

**Table 3 nanomaterials-12-04298-t003:** Summary of features of achromatic metalenses in the visible light spectrum.

Refs	*NA*	∆λ (nm)	Focusing Efficiency	Notes
This study	0.33	470–700	Average: 31.71%	Polarization-insensitive
Ref. [23]	0.075	460–700	Maximum: 35%	Polarization-sensitive
Ref. [31]	0.2	470–670	500 nm: 20%	Polarization-sensitive
Ref. [33]	0.013	500–550	-	Polarization-insensitive
Ref. [35]	0.215717	400–660	Average: 33.6%	Polarization-insensitive

## Data Availability

The data that support the plots within this paper and other findings of this study are available from the corresponding authors upon reasonable request.

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
