# Peer review of "Broadband Achromatic Metalens in the Visible Light Spectrum Based on Fresnel Zone Spatial Multiplexing"

_nanomaterials, 2022, doi:10.3390/nano12234298_

Round 1
Reviewer 1 Report
- • The paper "Broadband achromatic metalens in the visible based on Fresnel band spatial multiplexing" written by Ruixue Shi, Shuling Hu, Chuanqi Sun, Bin Wang and Qingzhong Cai is dedicated to theoretical description of achromatic metalens which has a similar focal length in all the visible spectral range. Such a metalens is an aim of many scientific investigations recent years, and of course it is very interesting to achieve this goal. However, I cannot recommend this article to be published at Nanomaterials.
- • The paper written inaccurately and contains a lot of grammar mistakes, it is hardly possible to list them all. More important, different parts of the paper are not matched with each other and do not describe proposed design well. Paper starts from the analysis of the phase changing which lens must provide and phase changes by different unit cells. Design of the lens which is based on this part of investigation is presented after 2.2 section which describe another physics. For reader it is a big confusion. Next, Fresnel Band Spatial Multiplexing is used incorrectly. Fresnel bands must have unequal radiuses, but authors put R_1 equal to R_2. Finally, discussion about C function is a big confusion too. Firstly, I don’t understand why authors use particle swarm algorithm to find the values of this function. Why is it not enough to find it as difference between “ideal” phase shift (which is known) and the real phase shift of the proposed design (which is known also)? Secondly, authors do not gave any information about how exactly this additional phase shift can be obtained.
- • But the most important thing that this article does not give scientific novelty. Using very the same algorithm, Capasso et al already obtained achromatic metalens more than five years ago [Khorasaninejad, M.; Shi, Z.; Zhu, A.Y.; Chen, W.T.; Sanjeev, V.; Zaidi, A.; Capasso, F. Achromatic metalens over 60 nm bandwidth in the visible and metalens with reverse chromatic dispersion. Nano Lett 2017, 17, 1819-1824.], and not only in simulations, but also experimentally. Authors can say that spectral band in this article is much smaller, but that’s because С function in experiment cannot have all the possible values as in theory. I want to stress that Capasso’s concept can be easily used in theory for metalens of any desirable spectral rang, so it is not the novelty of this paper.
- • I don’t see the sense to list less important mistakes and inaccuracies besides of the main issues discussed above.
Author Response
Response to Reviewer Comments
We appreciate your professional guidance and advice. In response to your guidance, we made a few changes to the article.
Point 1: The paper written inaccurately and contains a lot of grammar mistakes, it is hardly possible to list them all.
Response 1: Thanks for your professional guidance. We have polished the article to make sure there are no grammatical mistakes.
Point 2: Design of the lens which is based on this part of investigation is presented after 2.2 section which describe another physics. For reader it is a big confusion.
Response 2: We are very grateful to your comments for the manuscript.We want to be clear that the metalen's time bandwidth product must be smaller than the phase library' time bandwidth product. Therefore, the time bandwidth product of the phase library is provided. If we know the time delay of the metalens, we can deduce the maximum bandwidth that the metalens can achieve. This can be used to check whether our phase library can meet our bandwidth requirements. That's why we wrote about the time bandwidth product in Section 2.2.
In order to make this section clearer, we have also made some adjustments in Section 2.2. Please see details on Page 6, lines 180-182.
Since the time bandwidth product is a crucial topic in optics, understanding the time bandwidth product of the metalens is essential. The maximum time bandwidth product of the metalens obtained in 2018[1] is 11.5 and that of Haoran Ren and Stefan A. Maier [2] is 6.01 in 2021.
[1]Presutt, F.; Monticone, F. Focusing on bandwidth: achromatic metalens limits. Optica 2020, 7(6), 624-631.
[2]Ren, H.; Jang, J.; Li, C.; Aigner, A.; Plidschun, M.; Kim, J.; Rho, J.; Schmidt, M.A.; Maier, S.A. An achromatic metafiber for focusing and imaging across the entire telecommunication range. Nat Commun 2022, 13, 4183.
Point 3: Fresnel Band Spatial Multiplexing is used incorrectly.
Response 3: We appreciate you fixing the Fresnel zone problem in our article. I apologize for the misconception that our writing caused. To be clear, rather than building two Fresnel lenses, we employed the idea of dividing the metalens into numerous Fresnel zones and positioning the 488nm and 632.8nm structures alternatively in each zone in our study. Region III belongs to the unit cells of 488nm, and Region IV belongs to the unit cells of 632.8nm. It is feasible that R3 equals R4.
Here are some references. Motivated by Yongkang Song's article [3], we considered the possibility of segmenting the lens into different zones along the radial direction in order to obtain achromatic function. First, we attempted to separate the metalens into two zones, however this was unsuccessful due to the presence of bi-foucs. We then sought to make it into four zones. We read Federico Cappso's article [4] at the same time, which inspired us and supported our conclusions. The team led by Capasso also divided the lens into multizones along the radial axis, but it employed reverse design to determine the number of zones and the radius. Then ,we set the zones’ range to 1:1:1:1, and the focal length at 632.8nm did not meet our requirements. Finally, as our metalens radius is 7 um, we set the region's range to 3:2:1:1. Finally, we set the region's range to 3:2:1:1 and used the particle swarm optimization algorithm to optimize.
Please see details on Page 2 lines 69-76, Page 6 lines 183-195 and the supplementary Figure S1.
- Song, Y.; Liu, W.; Wang, X.; Wang, F.; Wei, Z.; Meng, H.; Lin, N.; Zhang, H. Multifunctional Metasurface Lens With Tunable Focus Based on Phase Transition Material. Frontiers in Physics, 2021, 9, 651898.
[4] Li, Z.; Lin, P.; Huang, Y. W.; Park, J. S.; Chen, W. T.; Shi, Z.; Qiu, C.; Cheng, J.; Capasso, F. Meta-optics achieves RGB-achromatic focusing for virtual reality. Science Advances 2021, 7(5), eabe4458.
Point 4: I don’t understand why authors use particle swarm algorithm to find the values of this function. Why is it not enough to find it as difference between “ideal” phase shift (which is known) and the real phase shift of the proposed design (which is known also)? Secondly, authors do not gave any information about how exactly this additional phase shift can be obtained.
Response 4: Thanks for your carefulness. Permit me to answer your questions on the additional phase. Here you suggest that the ideal phase and the actual phase have already been established and do not require the particle swarm optimization algorithm to determine them. What we want to rectify here is that we merely offered the additional phase in Equation (1) and did not give the value of the additional phase. Instead, to find the ideal additional phase, we made use of the particle swarm optimization algorithm. This is why you do not know where the additional phase occurs. In Figure 4, we have given a diagram of how to find C(λ). The particle is the additional phase and the wave difference is the fitness function. To reduce the wave difference, we continue iteratively updating additional phases.
Our working logic is:
- Divide the hyperlens into Four zones(3:2:1:1).
- Initialize the particle swarm parameter the additional phases of 488nm and 632.8nm are randomly given by the position of the particle.
- Calculate the ideal phase of 488nm and 632.8nm by Equation (1).
- Alternately arrange the unit cell of 488nm and 632.8nm into the four zones.
- Obtain the realized phase by IV.
- Calculate the phase dieviation as fitness vaule.
- Check whether the stop condition is met.
- If the stop condition is not met, update the particle swarm parameters and repeat steps III-VIII.
- If the stop condition is reached, end the iteration.
We modified the flow in Figure 4 to make it look clearer. Please see the page 8.
Point 5: But the most important thing that this article does not give scientific novelty. Using very the same algorithm, Capasso et al already obtained achromatic metalens more than five years ago [Khorasaninejad, M.; Shi, Z.; Zhu, A.Y.; Chen, W.T.; Sanjeev, V.; Zaidi, A.; Capasso, F. Achromatic metalens over 60 nm bandwidth in the visible and metalens with reverse chromatic dispersion. Nano Lett 2017, 17, 1819-1824.]
Response 5: We are very grateful to your comments for the manuscript. Let me answer your question about particle swarm optimization. We are appreciative to the article that capasso published in 2017 you mentioned because it has been a terrific resource for us.The paper helped us a lot, and we've already put it in the reference[36] in our article. Again, we want to make it clear that we're not just using PSO for achromatic change, and that's not our main innovation. Our innovation is to divide the metalens into multiple zones, and then reduce the wave difference by particle swarm optimization.
In 2017, Khorasaninejad, M, Capasso, F. et al. have used the particle swarm optimization algorithm to design achromatic metalens, of course, you also made it clear that their bandwidth is only 60nm [4]. Following Capasso's 2017 proposal of the particle swarm optimization algorithm, in 2018, Capasso's team proposed the technique for adjusting group delay and group delay dispersion to produce achromatic metalens[5].Possibly one of the reasons is that their original approach was relatively limited. In the same year, a different approach to achromatic metalens design was developed by the Shuming Wang’ team[6]. In 2021, Capasso's group proposed employing multizones and dispersion engineering to develop an achromatic metalens[7].
Up to now, particle swarm optimization is still widely used in the design of metalens. In 2021, Liang Yu et al. realized achromatic 500-550nm by using particle swarm optimization algorithm[8]. In 2022, Long Chen et al. realized the dual focus achromatic design at 633 nm, 532nm and 473nm[9].
So far, the realization of metalens with large bandwidth, high numerical aperture and high focusing efficiency in the range of visible light band has been a challenging and interesting research topic.
- Khorasaninejad, M.; Shi, Z.; Zhu, A.Y.; Chen, W.T.; Sanjeev, V.; Zaidi, A.; Capasso, F. Achromatic metalens over 60 nm bandwidth in the visible and metalens with reverse chromatic dispersion. Nano Lett 2017, 17, 1819-1824.
- Chen, W.T.; Zhu, A.Y.; Sanjeev, V.; Khorasaninejad, M.; Shi, Z.; Capasso, F. A broadband achromatic metalens for focusing and imaging in the visible. Nature Nanotechnology 2018, 13, 220-226.
- Wang, S.; Wu, P.C.; Su, V.C.; et al. A broadband achromatic metalens in the visible. Nature nanotechnology 2018, 13(3), 227-232.
- Li, Z.; Lin, P.; Huang, Y. W.; Park, J. S.; Chen, W. T.; Shi, Z.; Qiu, C.; Cheng, J.; Capasso, F. Meta-optics achieves RGB-achromatic focusing for virtual reality. Science Advances 2021, 7(5), eabe4458.
- Liang, Y.; Xu, Y.; Yang, Z.; Xue, S.; Liao, J. Pan, Y.; Wang, Y. Design and research of polarization-free achromatic metalens. Chinese Laser 2021, 48, 0303001.
- Chen, L.; Shao, Z.; Liu, J.; et al. Multi-wavelength achromatic bifocal metalenses with controllable polarization-dependent functions for switchable focusing intensity. Journal of Physics D: Applied Physics 2021, 55(11), 115102.
Finally, we would like to express our heartfelt gratitude for your advice and recommendations, which helped us improve our article.

Reviewer 2 Report
This manuscript introduces a metalens that achieves fair focussing for at least ten wavelengths over the visible spectrum. That is interesting. It is based on particle swarm optimization of nanopillars, all of the same height, and distributed on a sampling grid that changes from one region of the pupil to another. There are four regions, alternately optimized for one of two specific wavelengths, 488 and 632.8 nm. A constant phase C(lambda) can be selected independently for each wavelength, which is a good optimization parameter.
It is claimed that the average "focusing efficiency" is 31.71% as defined in section 3.1. Whether this is appropriate for some applications is an open question.
I believe that this work is original as summarized above and therefore it can be published in an archival journal such as Nanomaterials. Before acceptance, I would request the authors to consider the following suggestions.
1) What defines an optimal sampling period for a given wavelength?
2) Line 100, how is the Nyquist theorem applied here?
3) Lines 100 and 103, there are two conflicting constraints on the nanopillar period, why is that? Are both needed?
4) Line 104, please explain why the period should be larger than the "wavelength band"
5) Line 105, why are those curves "resonance" curves?
6) Line 64: why does it make fabrication easier if all nanostructures have the same "shape", if they do not all have the same diameter? To make my point clear, is the difficulty for fabricating a 10 nm diameter 1.3 micrometer height pillar the same as for fabricating a 1 micrometer diameter 1.3 micrometer height pillar?
7) Has the tolerance to incidence angle been tested?
8) Has the tolerance to minute fabrication defects such as dispersion in the nanopillar size been tested?
9) Line 145: how have the number of regions and their respective sizes been selected?
10) Line 149, I do not understand why there is no cross-coupling.
Author Response
We appreciate your professional guidance and advice. In response to your guidance, we made a few changes to the article. Please see the attachment

Reviewer 3 Report
The authors present a broadband achromatic metalens in the visible relied on Fresnel band spatial multiplexing. The novelty of this manuscript is quite clearly presented. The authors combined the Fresnel band spatial multiplexing method and the particle swarm algorithm to correct the chromatic aberration of the metalens. However, some points can be considered to further improve the manuscript. My detailed feedback can be found below:
Introduction
1. In the first paragraph, the author mentioned “Metalenses focus the light at a certain point through continuous accumulation of optical paths.” I am quite confused about this comment. Because metalens is a flat lens technology made by optical components that use metasurfaces. However, the metasurface can control the phase of the light by introducing an abrupt phase shift. Can you explain more clearly in your comment?
2. In the last paragraph, the authors mentioned “To our knowledge, few designs apply Fresnel band multiplexing to achieve broadband achromatic metalenses”. Can you add some references for this comment?
Method
The method section is described in quite a detail. However, some important things need to be clear, for example:
3. Why did the authors choose “the ratio of radii R1, R2, R3, and R4 of four regions is 3:2:1:1”?
4. How many unit cells did the authors use for each metalens for each region?
Results
5. The authors should list the focal length of metalens at 12 discrete wavelengths using a table instead of only mentioning the equation of focal length change.
6. The authors summarized the key parameters of the proposed metalens in table 1. However, it is better when the authors should make a table comparing the results of the proposed metalens with previous studies in the visible band.
Author Response
Response to Reviewer Comments
We appreciate your professional guidance and advice. In response to your guidance, we made a few changes to the article. You can see the attachment.
Introduction
Point 1: In the first paragraph, the author mentioned “Metalenses focus the light at a certain point through continuous accumulation of optical paths.” I am quite confused about this comment. Because metalens is a flat lens technology made by optical components that use metasurfaces. However, the metasurface can control the phase of the light by introducing an abrupt phase shift. Can you explain more clearly in your comment?
Response 1: We are very grateful to your comments for the manuscript. Based on the equivalent medium theory, each meta-unit can be considered as a truncated waveguide[1 ,2]. light is concentrated inside the meta-units that behave as weakly coupled low-quality factor resonators[1, 2]. The final amplitude and phase of light is the superposition of optical paths. The meta unit acts like a resonator and offer the phase shift [3]. When the electromagnetic wave passes through the metalens, it controls the outgoing phase of the electromagnetic wave by modulating the accumulated phase of the electromagnetic wave on the metalens. The phase of light generated by the meta-unit with height H is
So, here we mentioned ”Metalenses focus the light at a certain point through continuous accumulation of optical paths”. To make it clearer, we have changed the sentence to ”Metalens focuses the light at a certain point via the phase shifters when the light scatters off the array of resonators comprising the metalens.”
Please see details on page 1, line 40-41 and the references[1-3] are also cited in the article.
[1]Khorasaninejad, M.; Shi, Z.; Zhu, A.Y.; Chen, W.T.; Sanjeev, V.; Zaidi, A.; Capasso, F. Achromatic metalens over 60 nm bandwidth in the visible and metalens with reverse chromatic dispersion. Nano Lett 2017, 17, 1819-1824.
[2]Khorasaninejad, M.; Capasso,F. Metalenses: Versatile multifunctional photonic components. Science 2017, 358(6367): eaam8100.
[3]Aiet, F.; Genevet, P.; Kats, M.A.; Yu, N.; Blanchard, R.; Gaburro, Z.; Capasso, F. Aberration-Free Ultrathin Flat Lenses and Axicons at Telecom Wavelengths Based on Plasmonic Metasurfaces. Nano Letters, 2012, 12(9): 4932-4936.
Point 2: In the last paragraph, the authors mentioned “To our knowledge, few designs apply Fresnel band multiplexing to achieve broadband achromatic metalenses”. Can you add some references for this comment?
Response 2: Thanks for your suggestions, we have added references to the article. In 2017, Ori Avayu used the well-known Fresnel binary zone plate configuration for three individual lenses[4]. In order to alleviate the chromatic aberrations of individual diffractive elements, Ori Avayu introduced dense vertical stacking of the three independent metasurfaces. In 2021, Yongkang Song et al divided the metalens into two zones to focus LCP and RCP[5]. In 2021, Federico Capasso et al achieved diffraction-limited achromatic focusing of the primary colors by exploiting constructive interference of light from multiple zones and dispersion engineering[6]. Meanwhile, other spatial multiplexing methods are used for discrete wavelength achromatic design.
Here we learn from the above idea of spatial reuse and realize the achromatic design of visible band with particle swarm optimization algorithm.
The above is written in the introduction and the references have been cited in the article. Please see details on page 2, lines 69-76.
[4]Avayu, O.; Almeida, E.; Prior, Y.; Ellenbogen, T.; Composite functional metasurfaces for multispectral achromatic optics. Nat Commun 2017, 8, 14992.
[5]Song, Y.; Liu, W.; Wang, X.; Wang, F.; Wei, Z.; Meng, H.; Lin, N.; Zhang, H. Multifunctional Metasurface Lens With Tunable Focus Based on Phase Transition Material. Frontiers in Physics 2021, 9(1): 651898.
[6]Li, Z.; Lin, P.; Huang, Y. W.; Park, J. S.; Chen, W. T.; Shi, Z.; Qiu, C.; Cheng, J.; Capasso, F. Meta-optics achieves RGB-achromatic focusing for virtual reality. Science Advances 2021, 7(5), eabe4458.
Method
Point 3: Why did the authors choose “the ratio of radii R1, R2, R3, and R4 of four regions is 3:2:1:1”?
Response 3: Motivated by Yongkang Song's article [3], we considered the possibility of segmenting the lens into different zones along the radial direction in order to obtain achromatic function. First, we attempted to separate the metalens into two zones, however this was unsuccessful due to the presence of bi-foucs. We then sought to make it into four zones. We read Federico Cappso's article [4] at the same time, which inspired us and supported our conclusions. The team led by Capasso also divided the lens into multizones along the radial axis, but it employed reverse design to determine the number of zones and the radius. Then ,we set the zones’ range to 1:1:1:1, and the focal length at 632.8nm did not meet our requirements. Finally, as our metalens radius is 7 um, we set the region's range to 3:2:1:1 and used the particle swarm optimization algorithm to optimize.
Please see details on page 6, lines 184-195 and supplementary Figure S1.
Point 4: How many unit cells did the authors use for each metalens for each region?
Response 4: In our design, the metalens has a diameter of 7 um and a period of 380 nm. So, there are 193, 360, 224 and 292 unit cells in Regions I, II, III and IIIV respectively. In Region I, II, III IV, there are 24, 19, 20, 9 unit cells of different diameters respectively.
Please see details on supplementary Table S1. We have listed the radius of the unit cells for different regions.
Results
Point 5: The authors should list the focal length of metalens at 12 discrete wavelengths using a table instead of only mentioning the equation of focal length change.
Response 5: Thank you for your guidance, we have put the table in the article. Please see details on page11, Table 1.
Point 6: The authors summarized the key parameters of the proposed metalens in table 1. However, it is better when the authors should make a table comparing the results of the proposed metalens with previous studies in the visible band.
Response 6: Thank you for your professional guidance. We have put the table in the article. Because different references give different key parameters, we have made a new table. Please see details on page 12, Table 3.
Finally, we would like to express our heartfelt gratitude for your advice and recommendations, which helped us improve our article.

Reviewer 4 Report
please see in pdf-file

Author Response
Response to Reviewer Comments
We appreciate your professional guidance and advice . In response to your guidance, we made a few changes to the article. You can see the attachment.
Point 1: Repeated words in the title: Table 1. Key parameters of the achromatic of the achromatic metalens. It is better to include this short (but very wide) Table in the text.
Response 1: Thank you.We have rewritten the title of Table 1. Please see details on page 11, Table 2.
Point 2: For more suitable formatting, the Fig. 5 should be given in 2 lines, but the Fig.7 – otherwise, in columns.
Response 2: Thanks for your professional guidance. We have reformatted Figure 5, 7. Please see details on page 10 and page 12.
Point 3: More tickmarks should be given in x-axes on Figs.5,7.
Response 3: Thanks for your carefulness. We have redrawn Figure 5,7 with more tickmarks in x-axes. Please see details on page 10 and page 12.
Finally, we would like to express our heartfelt gratitude for your advice and guadiance, which helped us improve our article.

Round 2
Reviewer 1 Report
I appreciate authors for step-by step answer and minor correcting of the manuscript. However, I did not receive answers to major issues and I can't recommend the article for publishing. Namely,
- I asked authors about expediency of particle swarm optimization. As an answer, authors described in details the optimization algorithm, but my actual question is: design of metalens is known (it was shown in Figure 3), real phase profile for any wavelength is known too. The desirable phase profile is also known (given by formula 1). Why is phase difference (equals to C) not known from these two facts? Authors added sentence in manuscript "The particle swarm optimization algorithm is used to identify the C(λ) corresponding to 488 nm and 632.8 nm, respectively and arrange the unit cells by zones multiplexing." I don't understand it. Was design of metalens corrected after optimization? How exactly? What is the final design of a metalens, if so?
- authors did not say anything about experimental ways to realize additional phase shifting, so it is still not clear how to build achromatic metalens in experiment.
- I did not understand explanation of lens zones' radiuses defining. Authors wrote that results for radiuses 3:2:1:1 are better than for radiuses 1:1:1:1, but it is pretty obvious and still did not explain physical origin well.
Author Response
Response to Reviewer Comments
We appreciate your professional guidance and advice. In response to your guidance, we made a few changes to the article. You can see the attachment.
Point 1: I asked authors about expediency of particle swarm optimization. As an answer, authors described in details the optimization algorithm, but my actual question is: design of metalens is known (it was shown in Figure 3), real phase profile for any wavelength is known too. The desirable phase profile is also known (given by formula 1). Why is phase difference (equals to C) not known from these two facts? Authors added sentence in manuscript "The particle swarm optimization algorithm is used to identify the C(λ) corresponding to 488 nm and 632.8 nm, respectively and arrange the unit cells by zones multiplexing." I don't understand it. Was design of metalens corrected after optimization? How exactly? What is the final design of a metalens, if so?
Response 1: Thank you for your professional guidance.
According to the guidance and questions, the characteristic of the particle swarm optimization(PSO) algorithm should be introduced at first. Particle swarm optimization is a finite number of dynamic iterative process, which has been used throughout the whole design process of metalens and the design of the metalens is constantly corrected in the whole process of PSO algorithm [1-3] .
In our article, Figure 4 gives the process of PSO, which clearly shows the dynamic recycling characteristic. The iteration of PSO algorithm is actually to find the minimum value of total deviation ∆. According to Equation 1, C(λ) is defiend as the additional phase shifting which is the key to realize broadband achromatic. In each iteration, C(λ) as the position of the particle swarm is updeated and optimized. According to Fresnel zones multiplexing, at each (x,y) of the metalens, the unit cell with the smallest deviation from the ideal phase is selected from phase library and the total deviation is calculated. When the ∆ satisfies the Rayleigh criterion ,the iteration stops. However, even if the deviation exceeds λ/4, when it is small enough, the wavefront can be considered defect-free[4]. In this paper, the number of iterations was defined to be 800. When the number of iterations reaches 800, our requirements have been met. C(λ) and the structure of metalens are finally determined.
In the end, after optimization, the additional phase corresponding to 488nm is 0.53628 and the additional phase corresponding to 632.8nm is -4.0775. In Region I, II, III IV, there are 24, 19, 20, 9 unit cells of different diameters respectively.
These references[1-4] have been already cited in line 192, line 231 (modified file : line 178, line 216) of our article. At the same time, we added the contents in our article. Please see details on Page 7, lines 218-219 and lines 222-227(modified file : Page 7, lines 202-203, lines 207-213).
We have listed the radius of the unit cells for different regions. Please see details on supplementary Table S1.
[1]Khorasaninejad, M.; Shi, Z.; Zhu, A.Y.; Chen, W.T.; Sanjeev, V.; Zaidi, A.; Capasso, F. Achromatic metalens over 60 nm bandwidth in the visible and metalens with reverse chromatic dispersion. Nano Lett 2017, 17, 1819-1824.
[2]Liu, M.;Xu, N.; Wang, B.; Qian, W.; Xuan, B.; Cao, J. Polarization independent and broadband achromatic metalens in ultraviolet spectrum[J]. Optics Communications, 2021, 497: 127182.
[3] Liang, Y.; Xu, Y.; Yang, Z.; Xue, S.; Liao, J. Pan, Y.; Wang, Y. Design and research of polarization-free achromatic metalens. Chinese Laser 2021, 48, 0303001.
[4]Barakat, R. Rayleigh wavefront criterion. Josa 1965, 55, 572-573.
Point 2: Authors did not say about experimental ways to realize additional phase shifting.
Response 2: Thank you for your carefulness.
According to the Equation 1, C(λ) is the additional phase shifting. C(λ) is constantly tuned and optimized throughout the whole design process of metalens by PSO algorithm. Since the PSO algorithm is a finite number of dynamic iterations, C(λ) is updated and optimized in each iteration until the last iteration C(λ) is determined. It is already answered in the response of Point 1 that the whole process is shown in Figure 4.
In the last iteration, the optimized constant C is obtained and the unit cell of the metalens at each position (x,y) is selected. In the end, after optimization, the additional phase corresponding to 488nm is 0.53628 and the additional phase corresponding to 632.8nm is -4.0775.
Point 3: I did not understand explanation of lens zones' radiuses defining. Authors wrote that results for radiuses 3:2:1:1 are better than for radiuses 1:1:1:1, but it is pretty obvious and still did not explain physical origin well.
Response 3: Thank you for your carefulness.
In Page 2 lines 67-74, we have introduced the referencens using Fresnel zones multiplexing. The electric field at the focal spot is determined by the interference of electric field within each zone and the interference of the electric fields from N different zones[5]. Therefore, it is necessary to reasonably select the number and the radius of Fresnel regions, as well as the unit cells of each region, so as to maximize the electric field intensity at the focal point. In this paper, we choose to try 1:1, 1:1:1:1, 3:2:1:1 three zones multiplexing and use particle swarm optimization algorithm to optimize. Finally we find that 3:2:1:1 multiplexing method meets our requirements.
We have added the contents in our article. Please see details on Page 6, lines 191-196(modified file : Page 6, lines 178-186).
[5]Li, Z.; Lin, P.; Huang, Y. W.; Park, J. S.; Chen, W. T.; Shi, Z.; Qiu, C.; Cheng, J.; Capasso, F. Meta-optics achieves RGB-achromatic focusing for virtual reality. Science Advances 2021, 7, eabe4458.
Finally, we would like to express our heartfelt gratitude for your advice and recommendations, which helped us improve our article.

Round 3
